# Dose response of running on blood biomarkers of wellness in generally healthy individuals

**Bartek Nogal**[1], **Svetlana Vinogradova**[1], **Milena Jorge**[1], **Ali Torkamani**[2,3], **Paul Fabian**[1], **Gil Blander**[1] *

1 InsideTracker, Cambridge, Massachusetts, United States of America, 2 The Scripps Translational Science Institute, The Scripps Research Institute, La Jolla, California, United States of America, 3 Department of Integrative Structural and Computational Biology, The Scripps Research Institute, La Jolla, California, United States of America

☯ These authors contributed equally to this work.

* gblander@insidetracker.com

**Data Availability Statement:** All Mendelian Randomization data required to replicate the causal analysis can be freely accessed at: https://gwas.mrcieu.ac.uk/. No special access is required and all

## Abstract

Exercise is effective toward delaying or preventing chronic disease, with a large body of evidence supporting its effectiveness. However, less is known about the specific healthspan-promoting effects of exercise on blood biomarkers in the disease-free population. In this work, we examine 23,237 generally healthy individuals who self-report varying weekly running volumes and compare them to 4,428 generally healthy sedentary individuals, as well as 82 professional endurance runners. We estimate the significance of differences among blood biomarkers for groups of increasing running levels using analysis of variance (ANOVA), adjusting for age, gender, and BMI. We attempt and add insight to our observational dataset analysis via two-sample Mendelian randomization (2S-MR) using large independent datasets. We find that self-reported running volume associates with biomarker signatures of improved wellness, with some serum markers apparently being principally modified by BMI, whereas others show a dose-effect with respect to running volume. We further detect hints of sexually dimorphic serum responses in oxygen transport and hormonal traits, and we also observe a tendency toward pronounced modifications in magnesium status in professional endurance athletes. Thus, our results further characterize blood biomarkers of exercise and metabolic health, particularly regarding dose-effect relationships, and better inform personalized advice for training and performance.

## Introduction

Physical inactivity is one of the leading modifiable behavioral causes of death in the US [1]. Worldwide, physical inactivity is estimated to account for about 8.3% of premature mortality, an effect size that is on the same order as smoking and obesity [2]. At the same time, the potent health benefits of exercise have been proven time and time again, with results so consistent across a wide variety of chronic diseases that some posit it can be considered a medical intervention [3–5]. However, since most investigators report the effects of exercise in either

the datasets used can be further freely accessed via the free "TwoSampleMR" R package as described in the methods (biomarker dataset codes are shared in Supplementary tables as well). The minimal dataset required to replicate the blood biomarker results has been uploaded as Supporting information.

**Funding:** InsideTracker was the sole funding source. The funder provided support in the form of salaries for authors B.N., S.V., P.F., M.J., and G.B., and was involved in the decision to publish, but did not have an impact on the experimental design, data analysis, and conclusions.

**Competing interests:** B.N., S.V., P.F., and G.B. are employees of InsideTracker. This does not alter our adherence to PLOS ONE policies on sharing data and materials.

diseased populations or athletes [6, 7], there exists a significant gap in knowledge as to the measurable effects of exercise in the generally healthy population who exercise for the purpose of improving their healthspan, which can be projected via established measures such as blood biomarkers [8–11].

It is well established that routine laboratory biomarkers are validated proxies of the state of an individual's overall metabolic health and other healthpan-related parameters [12]. A large body of evidence supports the effectiveness of exercise in modifying blood biomarkers toward disease mitigation in clinical cohorts as well as athletes, where the effect sizes may be larger [6, 13]. Indeed, it's been shown that more favorable changes in response to exercise training occur usually in those with more pronounced dyslipidemia [13]. In professional athletes, the sheer volume and/or intensity of physical activity may drive large effects in various hematological, lipid, immune, and endocrine variables [6]. Our aim is to help fill the gap in understanding of the effects of exercise on blood biomarkers in the generally healthy, free-living population. Toward this end, we endeavored to explore the effects of vigorous exercise such as running in apparently healthy, mostly non-athletic cohort to better understand the landscape of blood biomarker modifications expected in the individual who partakes in recreational physical activity for the purpose of maintaining good health.

For this purpose, we leveraged the InsideTracker dataset that includes information on self-reported exercise habits combined with blood biomarker and genomics data. We have previously reported on the results of a longitudinal analysis on blood biomarker data from 1032 generally healthy individuals who used our automated, web-based personalized nutrition and lifestyle platform [14]. For the purpose of this investigation, we focused on running as the exercise of choice as it is one of the most common (purposeful) physical activity modalities practiced globally by generally healthy individuals and would thus be relevant. Moreover, since this was a cross-sectional study based on self-reported exercise habits, we attempted to increase our capacity to infer intervention effects, as well as tease out potential confounders, by performing 2S-MR in large independent cohorts.

## Materials and methods

### Dataset

We conducted an observational analysis of data from InsideTracker users. InsideTracker is a direct-to-consumer (DTC) company established in 2009 that markets and sells InsideTracker (insidetracker.com), a personalized lifestyle recommendation platform. The platform provides serum biomarker and genomics testing, and performs integrative analysis of these datasets, combined with activity/sleep tracker data toward biomarker and healthspan optimization (of note, at the time of this analysis, we did not have sufficient users with activity/sleep tracker data to include this data stream in the current study). New users were continuously added to the InsideTracker database from January 2011 to March 2022.

### Recruitment of participants

Recruitment of participants aged between 18 and 65 and residing in North America was conducted through company marketing and outreach. Participants were subscribing members to the InsideTracker platform and provided informed consent to have their blood test data and self-reported information used in an anonymized fashion for research purposes. Research was conducted according to guidelines for observational research in tissue samples from human subjects. Eligible participants completed a questionnaire that included age, ethnicity, sex, dietary preferences, physical activity, and other variables. This study employed data from 23,237 participants that met our analysis inclusion requirements, namely absence of any chronic

disease as determined by questionnaire and metabolic blood biomarkers within normal clinical reference ranges. The platform is not a medical service and does not diagnose or treat medical conditions, so medical history and medication use were not collected. The Institutional Review Board (IRB) determine this work was not subject to a review based on category 4 exemption ("secondary research" with de-identified subjects).

### Biomarker collection and analysis

Blood samples were collected and analyzed by Clinical Laboratory Improvement Amendments (CLIA)–approved, third-party clinical labs (primarily Quest Diagnostics and LabCorp). Participants were instructed to fast for 12 hours prior to the phlebotomy, with the exception of water consumption. Results from the blood analysis were then uploaded to the platform via electronic integration with the CLIA-approved lab. Participants chose a specific blood panel from 7 possible offerings, each comprising some subset of the biomarkers available. Due to the variation in blood panels offered, the participant sample size per biomarker is not uniform.

### Biomarker dataset preparation

In our raw dataset, occasional outlier values were observed that were deemed implausible (e.g. fasting glucose < 65 mg/dL). To remove anomalous outliers in a systematic way, we used the Interquartile Range (IQR) method of identifying outliers, removing data points which fell below Q1–1.5 IQR or above Q3 + 1.5 IQR. The cohort was divided into five groups: professional endurance runners (PRO), high volume amateur (>10 h/week, HVAM), medium volume amateur (3–10 h/week, MVAM), low volume amateur (<3 h/week, LVAM), and sedentary (SED).

### Calculation of polygenic scores

The variants (SNPs) comprising the polygenic risk scores were derived from publicly available GWAS summary statistics (https://www.ebi.ac.uk/gwas/). Scores were calculated across users by summing the product of effect allele doses weighted by the beta coefficient for each SNP, as reported in the GWAS summary statistics. Variant p-value thresholds were generally chosen based on optimization of respective PGS-blood biomarker correlation in the entire Inside-Tracker cohort with both blood and genomics datasets (~1000–1500 depending on the blood biomarker at the time of analysis). Genotyping data was derived from a combination of a custom InsideTracker array and third party arrays such as 23andMe and Ancestry. Not all variants for any particular PGS were genotyped on every array; proxies for missing SNPs were extracted via the "LDlinkR" package using the Utah Residents (CEPH) with Northern and Western European ancestry (CEU) population (R2 > 0.8 cut-off). Only results PGSs for which there was sufficient biomarker-genotyping dataset overlap were reported (note that none of the blood biomarker PGSs met this requirement).

### Blood biomarker analysis with respect to running volume and polygenic scores

To estimate significance of differences for blood biomarkers levels among exercise groups, we performed 3-way analysis of variance (ANOVA) analysis adjusting for age, gender, and BMI (type-II analysis-of-variance tables function ANOVA from 'car' R package, version 3.0–12). When estimating the effort of reported training volume on biomarkers, we assigned numerical values corresponding to 4 levels of running and performed ANOVA analysis with those levels treating it as an independent variable. P-values were adjusted using the Benjamini & Hochberg

method [15]. P-values for interaction plots were calculated with ANOVA including interaction between exercise group and polygenic scores category. When comparing runners (PRO and HVAM combined) versus sedentary individuals, we used propensity score matching method to account for existing covariates (age and gender): we identified 745 sedentary individuals with similar to runners' age distributions among both males and females. We used 'MatchIt' R package (version 4.3.3) implementing nearest neighbor method for matching [16].

## Mendelian randomization

We attempted to add insight around the causality of exercise vs. BMI differences with respect to serum marker improvement by performing MR analyses on a subset of biomarker observations where BMI featured as a strong covariate and was thus used as the IV in the 2S-MR. Thus, our hypothesis here was that BMI differences were the primary (causal) driver behind the improvement behind some biomarkers. MR uses genetic variants as modifiable exposure (risk factor) proxies to evaluate causal relationships in observational data while reducing the effects of confounders and reverse causation (S1 Fig in S1 File). These SNPs are used as instrumental variables and must meet 3 basic assumptions: (1) they must be robustly associated with the exposure; (2) they must exert their effect on outcome via the exposure, and (3) there must be no unmeasured confounders of the associations between the genetic variants and outcome (e.g. horizontal pleiotropy) [17]. Importantly, SNPs are proper randomization instruments because they are determined at birth and thus serve as proxies of long-term exposures and cannot, in general, be modified by the environment. If the 3 above mentioned assumptions hold, MR-estimate effects of exposure on outcomes are not likely to be significantly affected by reverse causation or confounding. In the 2S-MR performed here, where GWAS summary statistics are used for both exposure and outcome from independent cohorts, reverse causation and horizontal pleiotropy can readily be assessed, and weak instrument bias and the likelihood of false positive findings are minimized as a result of the much larger samples sizes [17]. Indeed, the bias in the 2S-MR using non-overlapping datasets as performed here is towards the null [17]. Furthermore, to maintain the SNP-exposure associations and linkage disequilibrium (LD) patterns in the non-overlapping populations we used GWAS datasets from the MR-Base platform that were derived from ancestrally similar populations ("ukb": analysis of UK Biobank phenotypes, and "ieu": GWAS summary datasets generated by many different European consortia). To perform the analysis we used the R package "TwoSampleMR" that combines the effects sizes of instruments on exposures with those on outcomes via a meta-analysis. We used "TwoSampleMR" package functions for allele harmonization between exposure and outcome datasets, proxy variant substitution when SNPs from exposure were not genotyped in the outcome data (Rsq>0.8 using the 1000G EUR reference data integrated into MR-Base), and clumping to prune instrument SNPs for LD (the R script used for MR analyses is available upon request). We used 5 different MR methods that were included as part of the "TwoSampleMR" package to control for bias inherent to any one technique [18]. For example, the multiplicative random effects inverse variance-weighted (IVW) method is a weighted regression of instrument-outcome effects on instrument-exposure effects with the intercept is set to zero. This method generates a causal estimate of the exposure trait on outcome traits by regressing the, for example, SNP-BMI trait association on the SNP-biomarker measure association, weighted by the inverse of the SNP-biomarker measure association, and constraining the intercept of this regression to zero. This constraint can result in unbalanced horizontal pleiotropy whereby the instruments influence the outcome through causal pathways distinct from that through the exposure (thus violating the second above-mentioned assumption). Such unbalanced horizontal pleiotropy distorts the association between the exposure and the

outcome, and the effect estimate from the IVW method can be exaggerated or attenuated. However, unbalanced horizontal pleiotropy can be readily assessed by the MR Egger method (via the MR Egger intercept), which provides a valid MR causal estimate that is adjusted for the presence of such directional pleiotropy, albeit at the cost of statistical efficiency. Finally, to ascertain the directionality of the various causal relationships examined, we also performed each MR analysis in reverse where possible.

## Results

### Study population characteristics

Table 1 shows the demographic characteristics of the study population. We observed a significant trend toward younger individuals reporting higher running volume, with more than 75% of the professional (PRO) group falling between the ages of 18 and 35 (S1 Table in S1 File). Significant differences were also observed in the distribution of males and females within study groups (Table 1). Moreover, higher running volume associated with significantly lower body mass index (BMI). Thus, moving forward, combined comparisons of blood biomarkers as they relate to running volume were adjusted for age, gender, and BMI.

### Endurance exercise exhibits a modest association with clusters of blood biomarker features

In order to begin to understand the most important variables that may associate with endurance exercise in the form of running, we performed a principal component analysis (PCA), sub-dividing the male and female cohorts into two most divergent groups in terms of exercise volume: PRO/high volume amateur (HVAM) and sedentary (SED) groups. Using propensity matching, PRO and amateur athletes who reported running >10h per week were combined into the PRO-HVAM group to balance out the sample size between the exercising and non-exercising groups. This approach yielded a modest degree of separation, with hematological, inflammation, and lipid features, as well as BMI explaining some of the variance (Fig 1A through 1D). We hypothesized that there may more subtle relationships between running volume and the blood biomarker features that contributed to distinguishing the endurance exercise and sedentary groups, thus we next performed ANOVA analyses stratified by running volume as categorized in Table 1.

### Significant trends in glycemic, hematological, blood lipid, and inflammatory serum traits with increasing running volumes

Weighted ANOVA analyses adjusted for age, gender, and BMI showed significant differences among groups for multiple blood biomarkers (Table 2 and S2 Table in S1 File, Figs 2 and 3).

**Table 1. Study population demographics.**

| Group | N | Female, % | Age, yrs | Body mass index, kg/m2 |
|---|---|---|---|---|
| PRO | 82 | 53.7% | 33.68 | 20.15 ± 6.02 |
| HVAM | 1103 | 52.9% | 39.48 | 22.57 ± 9.97 |
| MVAM | 6747 | 54.2% | 41.49 | 23.35 ± 9.76 |
| LVAM | 10877 | 34.2% | 41.16 | 24.72 ± 9.70 |
| SED | 4428 | 48.9% | 44.25 | 27.83 ± 10.70 |

PRO = Professional, HVAM = high volume amateur (>10 h/week), MVAM = medium volume amateur (3–10 h/week), LVAM = low volume amateur (<3 h/week), SED = sedentary

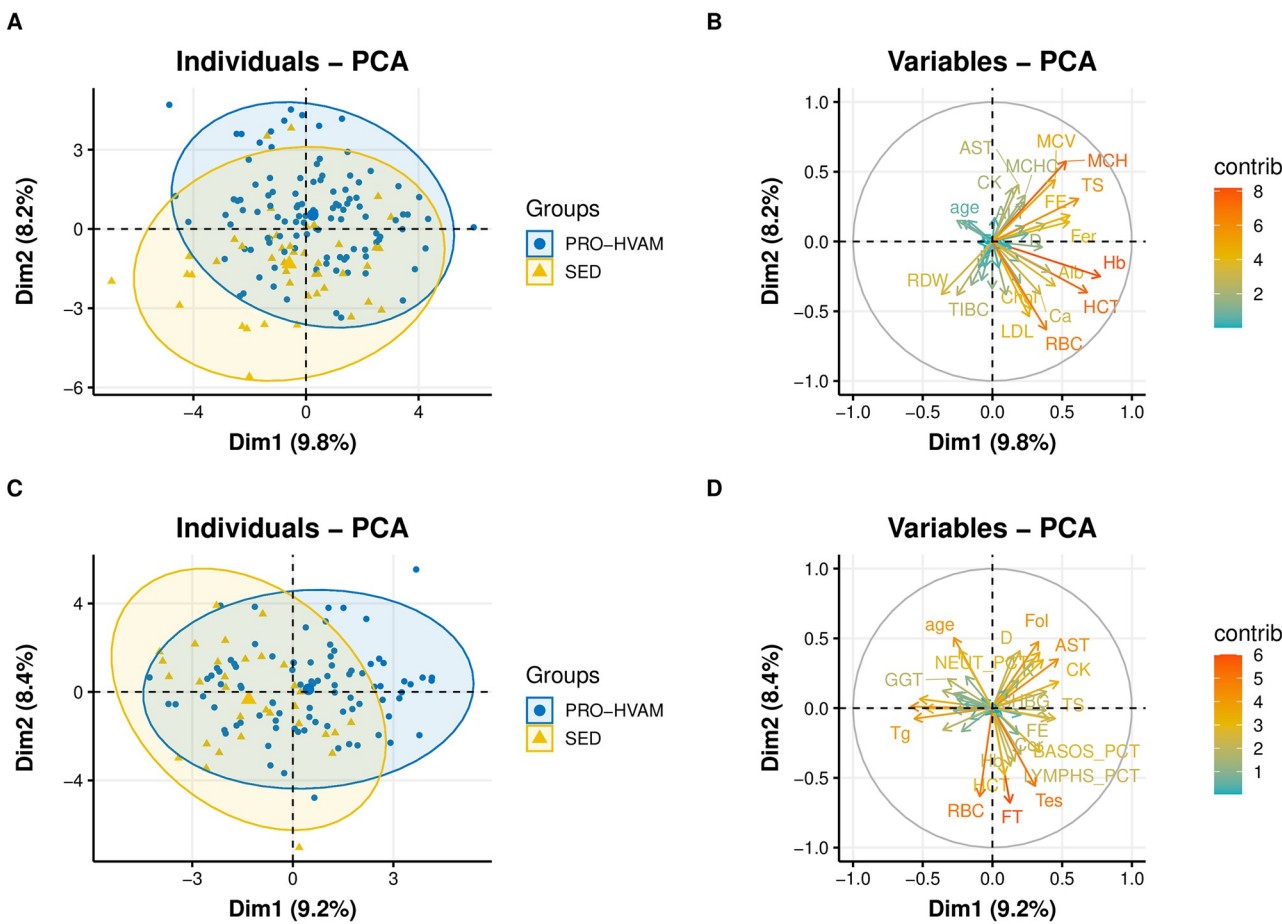

**Fig 1. Principal component analysis and variables plots of PRO-HVAM runners and sedentary user blood biomarkers.** Females, (A) and (B); males (C) and (D). PRO-HVAM = combined professional and high-volume amateur. Alb = albumin, ALT = alanine transaminase, AST = aspartate aminotransferase, B12 = vitamin B12, Ca = calcium, Chol = total cholesterol, CK = creatine kinase, Cor = cortisol, FE = iron, EOS_PCT = eosinophil percentage, Fer = ferritin, Fol = folate, FT = free testosterone, GGT = gamma-glutamyl transferase, Glu = glucose, Hb = hemoglobin, HCT = hematocrit, HDL = high density lipoprotein, HbA1c = glycated hemoglobin, hsCRP = high-sensitivity C-reactive protein, LDL = low density lipoprotein, LYMPS_PCT = lymphocyte percentage, MCH = mean cell hemoglobin, Mg = magnesium, MONOS_PCT = monocytes percentage, MPV = mean platelet volume, Na = sodium, RBC = red blood cells, RBC_Mg = red blood cell magnesium, RDW = red blood cell distribution width, SHBG = sex hormone binding globulin, Tg = triglycerides, TIBC = total iron binding capacity, WBC = white blood cells.

We observed a trend toward lower HbA1c, hsCRP, RDW, WBC, ferritin, gamma-glutamyl transferase (GGT), and LDL. HDL, hemoglobin (Hb), transferrin saturation (TS), alanine aminotransferase (ALT), aspartate aminotransferase (AST), vitamin B12, folate, 25-hydroxy vitamin D, and creatine kinase (CK) tended to be higher with increasing reported training volume, particularly in PRO runners (Table 2 and S2 Table in S1 File, Figs 2 and 3 and S2 Fig in S1 File). Hct and Hb were higher only in PRO males, whereas increased running volume associated with upward trend in these biomarkers in females (Fig 3A and 3B). Increased running volume was associated with markedly lower Fer in males, whereas female runners did not exhibit varying levels, and SED females showed increased levels (Fig 3C). The low ferritin observed in male and female runners was not clinically significant. ALT positively associated with running volume in females only (S2 Fig in S1 File). Serum and RBC magnesium (Mg) were both significantly lower in PRO runners relative to all other groups (Table 2 and Fig 3D

**Table 2. Blood biomarkers significantly different among sedentary individuals and those who partake in running for exercise to various degrees.**

| BIOMARKER | ANOVA P-VALUE | TREND P-VALUE | LOWEST MEAN | HIGHEST MEAN |
|-----------|---------------|---------------|-------------|--------------|
| ALB | <1e-16 | **<0.001** | MVAM | PRO |
| ALT | <1e-16 | **<1e-16** | SED | PRO |
| AST | <1e-16 | **<0.001** | SED | PRO |
| B12 | <0.001 | **<0.001** | SED | PRO |
| CHOL | <0.001 | **0.005** | PRO | SED |
| CK | <1e-16 | **<1e-16** | SED | PRO |
| COR | <0.001 | **0.675** | SED | PRO |
| FE | <0.001 | 0.119 | SED | PRO |
| FER | <1e-16 | **<1e-16** | MVAM | SED |
| FOL | <1e-16 | **<0.001** | SED | PRO |
| FT | <0.001 | **0.013** | SED | PRO |
| GGT | <1e-16 | **<0.001** | PRO | SED |
| GLU | 0.087 | 0.184 | PRO | SED |
| HB | 0.002 | **<0.001** | MVAM | PRO |
| HCT | 0.053 | 0.055 | MVAM | PRO |
| HDL | <1e-16 | **<0.001** | SED | PRO |
| HBA1C | <0.001 | **0.010** | PRO | SED |
| HSCRP | <0.001 | **0.176** | PRO | SED |
| LDL | <0.001 | **0.006** | PRO | SED |
| MG | <0.001 | 0.276 | PRO | SED |
| MPV | 0.058 | 0.089 | SED | HVAM |
| NA | <1e-16 | 0.622 | HVAM | SED |
| RBC_MG | <0.001 | 0.773 | PRO | SED |
| RDW | <1e-16 | **0.002** | PRO | SED |
| SHBG | <1e-16 | **0.004** | SED | PRO |
| TG | <1e-16 | **<1e-16** | PRO | SED |
| WBC | <1e-16 | **<1e-16** | PRO | SED |

and 3E). Increasing levels of endurance exercise also appeared to be associated with higher sex-hormone binding globulin (SHBG), particularly in PRO male runners (Fig 3F).

### Endurance exercise correlates with lower BMI across categories of genetic risk

Using publicly available GWAS summary statistics, we constructed blood biomarker polygenic risk scores (PGSs) to explore potential genetic risk-mitigating effects of endurance exercise. Since only a subset of the individuals in our cohort were genotyped, we aggregated the groups into 2 categories—PRO-HVAM and sedentary—to increase statistical power. This across-group sample size increase generally did not sufficiently power the ANOVA analysis to detect statistically significant trends, though the BMI polygenic risk was suggestively mitigated for both males and female PRO-HVAM runners across categories of genetic risk (Fig 4B).

### Increased running volume is associated with lower BMI which may drive biomarker changes

We observed a significant downward trend in the BMI with increased running volume for both males and females, and, although some of the biomarker differences between sedentary

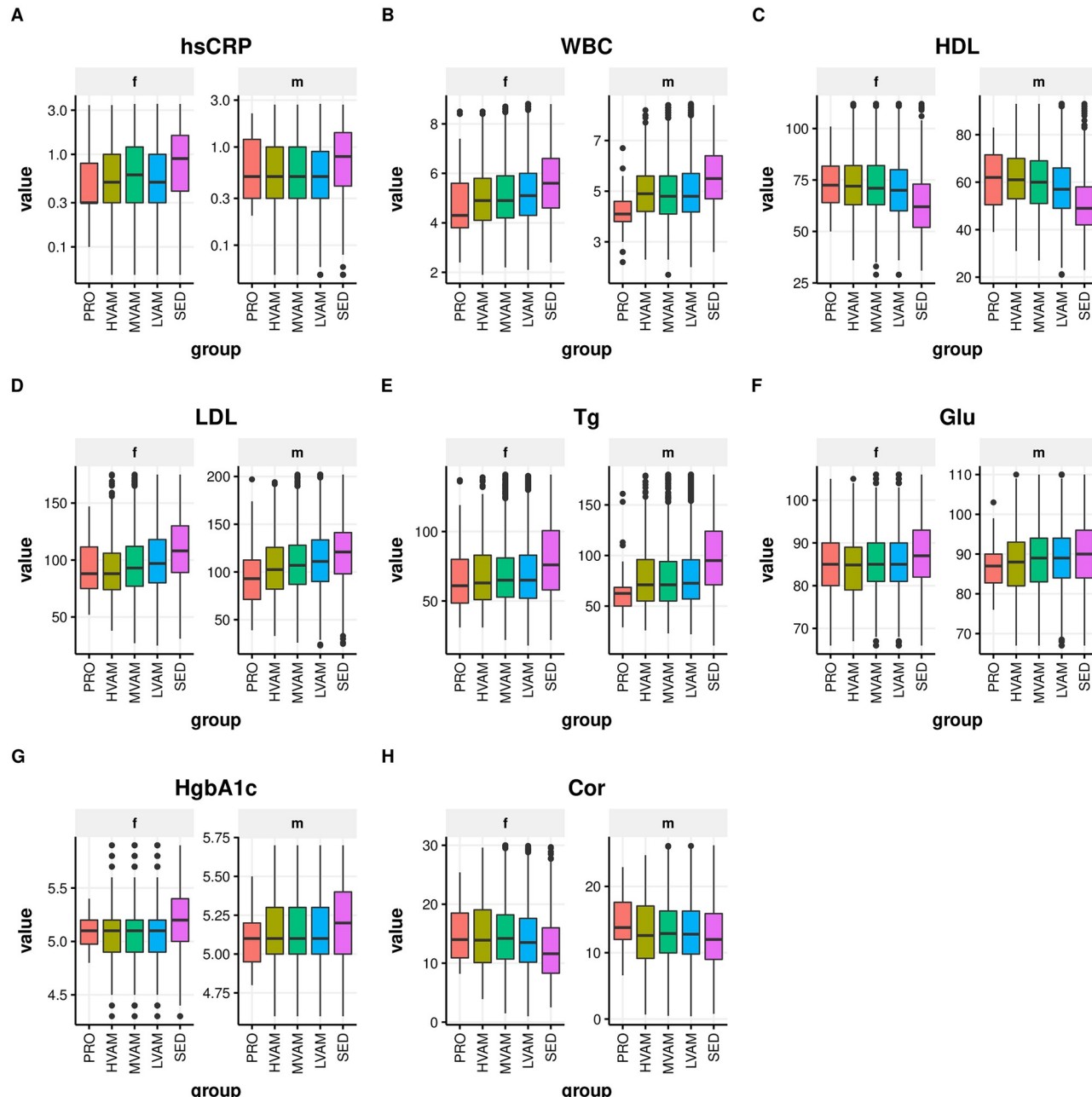

**Fig 2.** Blood biomarkers associated with running: Inflammation proxies, (A) hsCRP = high-sensitivity C-reactive protein and (B) WBC = white blood cells; blood lipids, (C) HDL = high density lipoprotein (D) LDL = low density lipoprotein, and (E) Tg = triglycerides; glycemia proxies, (F) Glu = glucose and (G) HgbA1c = glycated hemoglobin, and (H) Cor = cortisol.

and exercising individuals remained significant after adjustment for BMI, their significance was attenuated (Fig 4A). Thus, we hypothesized that BMI may be driving a significant portion of the observed variance in some of the biomarkers across the groups. Thus, to explore causal relationships between weight and biomarker changes, we performed 2S-MR with BMI-associated single-nucleotide polymorphisms (SNPs) as the instrumental variables (IVs) for a subset of the healthspan-related biomarkers where BMI explained a relatively large portion of the

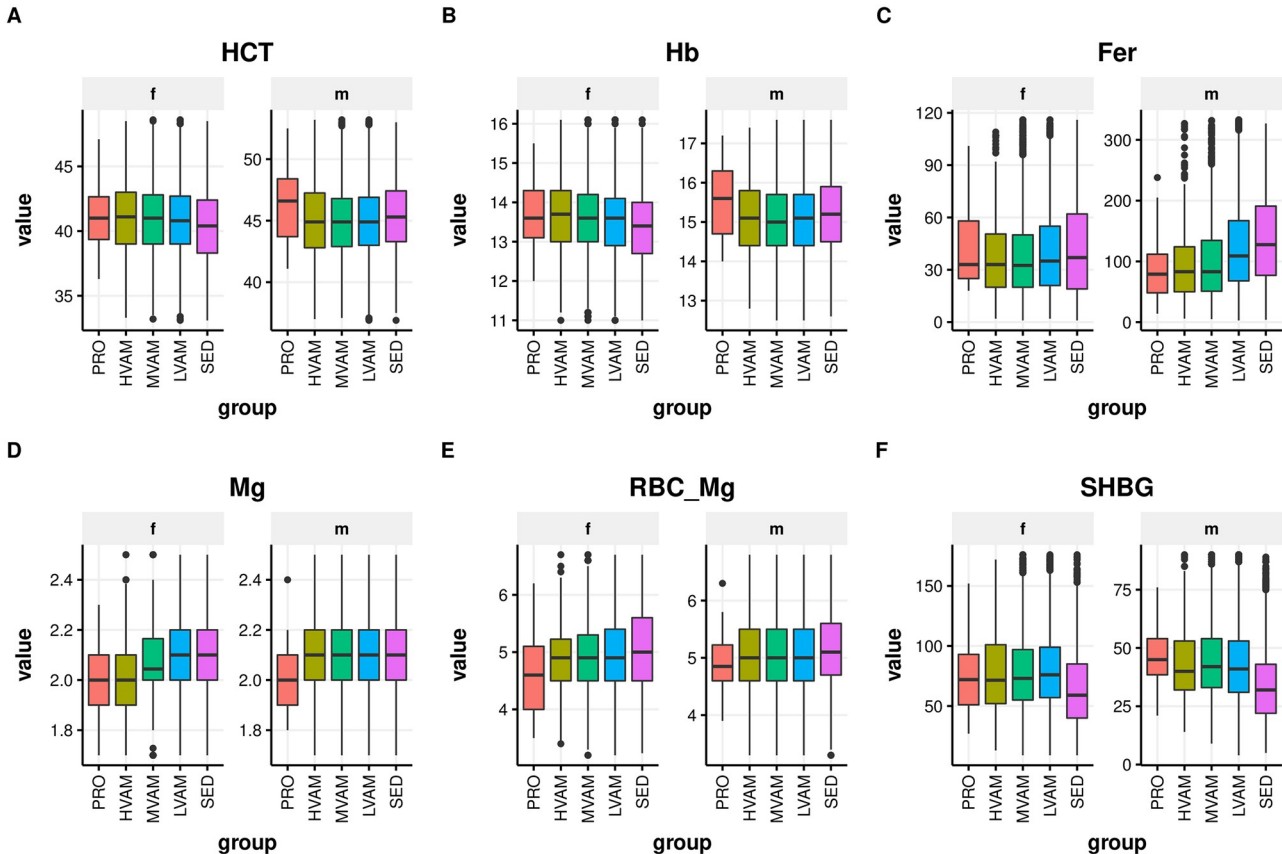

**Fig 3.** Blood biomarkers associated with running: (A and B) Hb (hemoglobin) and Hct (hematocrit) increase with increasing running volume, (C) Fer (ferritin) is reduced with increasing running volume, (D and E) Serum and RBC Mg (red blood cell magnesium) are reduced in professional runners, and (F) SHBG (sex hormone binding globulin) levels increase with increasing running volume in males.

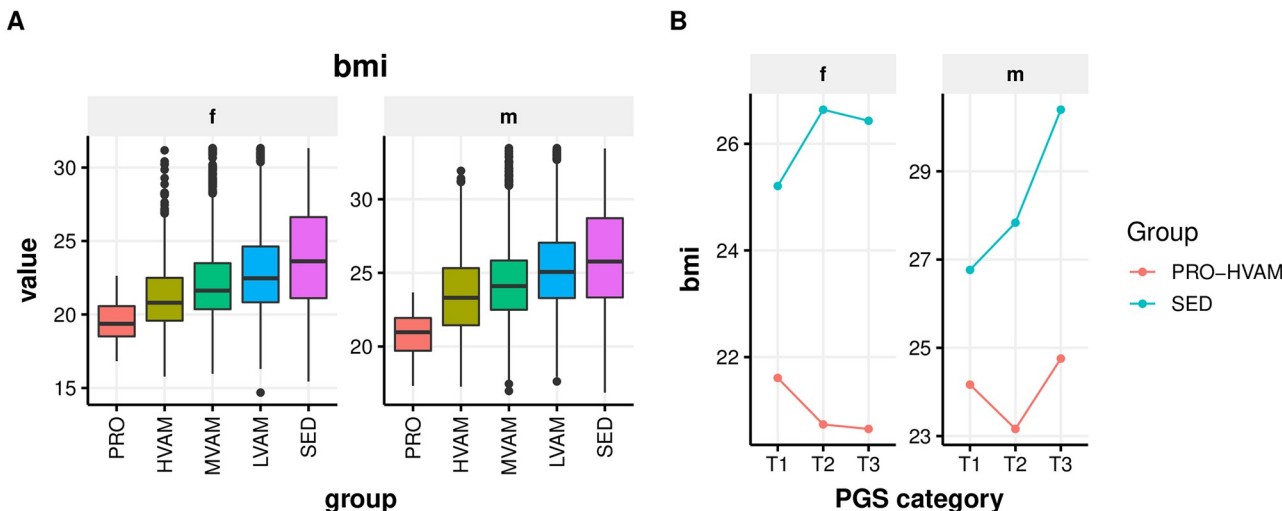

**Fig 4. BMI significantly varied among running groups (A) with some suggestive effects on BMI PGS modification (total number for observations (N) for T1, T2, and T3 were 87, 84, and 100, respectively) (B) T1, T2, and T3 = 1st, and 2nd and 3rd tertials of the polygenic score distribution.**

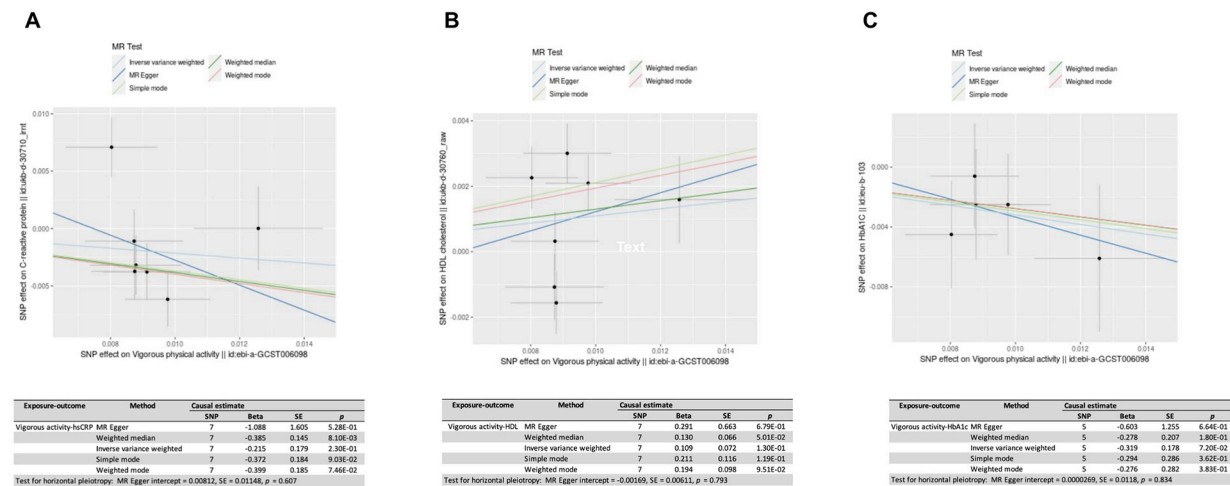

**Fig 5. Two-sample Mendelian randomization shows that increasing levels of vigorous physical activity such as running is associated with improvement of (A) hsCRP = high-sensitivity C-reactive protein, (B) HDL = high density lipoprotein, and (C) HbA1c = glycated hemoglobin levels.**

variance in our analysis. In general, these blood biomarkers associated with inflammation (hsCRP and RDW), lipid metabolism (Tg and HDL), glycemic control (HbA1c and Glu), as well as Alb and SHBG. We used GWAS summary statistics and found that most of these BMI-blood biomarker relationships examined directionally aligned with our study (except for LDL), and some were indicative of causal relationships in the BMI-biomarker direction even after considering directional pleiotropy (S3 Table in S1 File). We entertained the possibility of reverse causality and thus repeated the 2S-MR using each of the biomarker levels as the exposure and BMI as the outcome, and the results were generally not significant (except for WBC–see S4 Table in S1 File). Of note, to estimate the direct causal effects of running on blood parameters, we attempted to find an instrumental variable for to approximate running as the exposure from publicly available GWAS summary statistics. Toward this end, we found that increasing levels of vigorous physical activity did associate with lower hsCRP, HbA1C, higher HDL, and possibly higher SHBG (although the explained variance ($R^2$) in this exposure was just 0.001009, the F statistic was 37.7, thus meeting the criteria of $F > 10$ for minimizing weak instrument bias) (Fig 5 and S3 Fig in S1 File; S5 Table in S1 File).

## Vigorous physical activity associates with healthier behaviors

We hypothesized that those who exercise regularly may also partake in other healthful lifestyle habits that may be contributing to more optimal blood biomarker signatures of wellness. However, our dataset did not allow for systematic accounting of other lifestyle habits across all running groups. Thus, we again leveraged the potential of the 2S-MR approach to inform potential confounding associations between modifiable exposures and found that vigorous physical activity such as running is at least suggestively associated with several behaviors associated with improved health (S4 Fig in S1 File). Our analysis showed that those who participate in increasing levels of vigorous physical activity may be less likely to eat processed meat (IVW p = 0.0000013), sweets (IVW p = 0.32), and nap during the day (IVW p = 0.13), while increasing their intake of oily fish (IVW p = 0.029), salad/raw vegetable intake (IVW p = 0.00016), and fresh fruit (IVW p = 0.0027) (S6 Table in S1 File). Furthermore, following our assessment

of reverse causality, we found evidence for the bidirectionality in the causal relationship between vigorous activity and napping during the day and salad/raw vegetable intake, perhaps suggesting some degree of confounding due to population stratification (S7 Table in S1 File). The suggestive positive effect of fresh fruit and processed meat intake on vigorous physical activity appeared to violate MR assumption (3) (S1 Fig in S1 File) (horizontal pleiotropy p-values 0.051 and 0.17, respectively–S5 Fig in S1 File).

## Discussion

In this report, we describe the variance in wellness-related blood biomarkers among self-reported recreational runners, PRO runners, and individuals who do not report any exercise. Overall, we find that 1) recreational running as an exercise appears to be an effective intervention toward modifying several biomarkers indicative of improved metabolic health, 2) an apparent dose-response relationship between running volume and BMI may itself be responsible for a proportion of the apparent metabolic benefits, and 3) both PRO-level status and gender appear to associate with heterogeneous physiological responses, particularly in iron and magnesium metabolism, as well as some hormonal traits.

### Self-reported running improves glycemia and lipidemia

We did not observe distinct clusters corresponding to self-reported high-volume/PRO runners and the sedentary upon dimension reduction. This is, perhaps, not unexpected due, in part, to the self-selected healthspan-oriented nature of our cohort, where even the sedentary subset of individuals tends to exhibit blood biomarker levels in the normal clinical reference ranges. Furthermore, the measurement of running volume via self-report may be vulnerable to overestimation, which may have contributed to the blending of sedentary and exercise groups with respect to the serum markers measured, resulting in only marginal separation between the groups [19, 20]. However, we did observe significant individual blood biomarker variance with respect to reported running volumes when the dataset was subjected to ANOVA, even after adjustment for age, sex, and BMI.

From among glycemic control blood biomarkers, we were able to detect a relatively small exercise effect in both fasting glucose and HbA1c in this generally healthy cohort, where the average measures of glycemia were below the prediabetic thresholds in even the sedentary subset of the cohort. Larger exercise intervention effects on metabolic biomarkers may be expected in cohorts that include individuals with more clinically significant baseline values [21].

Similarly, blood lipids improved with higher self-reported running volume, and this result has been reported before in multiple controlled endurance exercise trials [22]. The literature indicates that HDL and Tg are two exercise-modifiable blood lipid biomarkers, with HDL being the most widely reported to be modified by aerobic exercise [23, 24]. Although the mechanism behind this is not entirely clear, it likely involves the modification of lecithin acyltransferase and lipoprotein lipase activities following exercise training [25]. We observed a similar trend in our blood biomarker analysis, with HDL exhibiting an upward trend with increasing reported running volume. While we also found Tg and LDL to decrease with increasing exercise volume, these trends were less pronounced. Reports generally suggest that, in order to reduce LDL more consistently, the intensity of aerobic exercise must be high enough [23]. In the case of Tg, baseline levels may have a significant impact on the exercise intervention effect, with individuals exhibiting higher baselines showing greater improvements [13].

Importantly, these results suggests that exercise has a significant effect on glycemic control and blood lipids even in the self-selected, already healthy individuals who are proactive about preventing cardiometabolic disease.

## Self-reported running and serum proxies of systemic inflammation

Chronic low-grade inflammation is one of the major risk factors for compromised cardiovascular health and metabolic syndrome (MetS). While there is no shortage of inflammation-reducing intervention studies on CVD patients with clinically high levels of metabolic inflammation, there is less emphasis on modifiable lifestyle factors that can help stave off CVD and extend healthspan in the generally healthy individual. Indeed, considering the pathological cardiovascular processes begin shortly after birth, prevention in asymptomatic individuals may be a more appropriate strategy toward decreasing the burden of CVD on the healthcare system [26].

Toward this end, increasing self-reported running volume appeared to associate with improved markers of inflammation, as shown by the lower levels of hsCRP, WBC, as well as ferritin. Of note, while the acute-phase protein, ferritin, is often used in the differential diagnosis of iron deficiency anemia, the biomarker's specificity appears to depend on the inflammatory state of the individual, as it associates with hsCRP and inflammation more than iron stores, particularly in those with higher BMI [27]. Although serum ferritin and iron is reported to be lower in male and female elite athletes [28], the observed overall negative association of ferritin with increased running volume in our cohort may be an indication of lower levels of inflammation rather than compromised iron stores, particularly since the average ferritin level across all groups was above the clinical iron deficiency thresholds. Moreover, increased levels of ferritin have been associated with insulin resistance and lower levels of adiponectin in the general population, both indicators of increased systemic inflammation [29]. Here, exercising groups with lower levels of ferritin also exhibited glycemic and blood lipid traits indicative of improved metabolic states, further supporting ferritin's role as an inflammation proxy. Finally, Hb, TS and iron tended to be higher in those who run for exercise compared to the SED group (with the TIBC lower), again suggesting that runners, including the PRO group, were iron-sufficient in this cohort.

## PRO endurance runners exhibit distinct biomarker signatures

PRO athletes exhibited lower serum and RBC Mg, which may be indication of the often-reported endurance athlete hypomagnesaemia [30]. While the serum Mg was still within normal clinical reference range for both PRO female and male athletes, RBC Mg, a more sensitive biomarker of Mg status [31], was borderline low in female PRO athletes and might suggest suboptimal dietary intakes and/or much higher volume of running training compared to the other running groups (i.e. >>10h /week). Indeed, this group also had elevated baseline CK and AST, which suggests a much higher training intensity and/or volume. Moreover, PRO level athletes had adequate iron status and serum B12 and folate in the upper quartile of the normal reference range, suggesting that these athletes' general nutrition status may have been adequate. These observations suggest that elite endurance runners may need to pay particular attention to their magnesium status.

Further, we observed higher levels of SHBG in PRO male runners, a biomarker whose levels positively correlate with various indexes of insulin sensitivity [32]. However, since the average SHBG levels in the SED group were not clinically low in both sexes, the observed increase in SHBG levels induced by running in males may be a catabolic response, as cortisol levels in this group were also higher. Indeed, Popovic et al. have shown that endurance exercise may

increase SHBG, cortisol, and total testosterone levels at the expense of free testosterone levels [33]. This could perhaps in part be explained by higher exercise-induced adiponectin levels, which have been shown to increase SHBG via cAMP kinase (AMPK) activation [34]. However, since our data is observational, we cannot rule out overall energy balance as a significant contributor to SHBG levels. For example, caloric restriction (CR) has been shown to result in higher SHBG and cortisol levels [32].

Finally, regarding the abovementioned PRO group elevated AST and CK biomarkers, evidence suggests that normal reference ranges in both CK and AST in well-recovered athletes should be adjusted up, as training and competition have a profound, non-pathological, impact on the activity of these enzymes [35, 36]. Indeed, the recommendation appears to be not to use reference intervals derived from the general population with hard-training (particularly competitive) athletes [36].

## Effect of BMI on blood biomarkers

Since the current study is a cross-sectional analysis of self-reported running, we could not rule out the possibility that factors other than exercise were the driving force behind the observed biomarker variance among the groups examined. These factors, such as diet, sleep, and/or medications were not readily ascertained in this free-living cohort at the time of this study, but BMI was readily available to evaluate this biomarker's potential relative contribution to the observed mean biomarker differences among self-reported runner groups.

Multiple studies have attempted to uncouple the effects of exercise and BMI reduction on blood biomarker outcomes, with mixed results [37]. For example, it is relatively well-known that acute bouts of exercise improve glucose metabolism, but long-term effects are less well described [38]. Indeed, whether exercise without significant weight-loss is effective toward preventing metabolic disease (and the associated blood biomarker changes) is inconclusive [39–41]. From the literature, it appears that, for endurance exercise to have significant effect on most blood biomarkers, the volume of exercise needs to be very high, and this typically results in significant reduction in weight. Thus, in practice, it is difficult to demonstrably uncouple the effects of significant exercise and the associated weight-loss, and the results may depend on the blood biomarker in question. Indeed, there is evidence that exercise without weight-loss does improve markers of insulin sensitivity but not chronic inflammation, with the latter apparently requiring a reduction in adiposity in the general population [39–41].

In our study of apparently healthy individuals, we observed a downward trend in BMI with increasing self-reported running volume, and, although this study was not longitudinal and we are thus unable to claim weight-loss, our 2S-MR analysis using BMI as the exposure nonetheless suggests this biomarker to be responsible for a significant proportion of the modification of some blood biomarkers.

**Serum markers of systemic inflammation.** Through our 2S-MR analyses, we show that BMI is causally associated with markers of systemic inflammation, including RDW, folate, and hsCRP [27, 42, 43]. Similar analyses have reported that genetic variants that associate with higher BMI were associated with higher CRP levels, but not the other way around [44]. The prevailing mechanism proposed to explain this relationship appears to be the pathological nature of overweight/obesity-driven adipose tissue that results in secretion of proinflammatory cytokines such as IL-6 and TNFa, which then stimulate an acute hepatic response, resulting in increased hsCRP levels (among other effects) [45]. Thus, our 2S-MR analyses and those of others [44] would indicate that the primary factor behind the lower systemic inflammation in our cohort may be the exercise-associated lower BMI and not running

exercise per se, though the lower hsCRP in runners remained significant after adjustment for BMI in our analysis.

Indeed, although a major driver behind reduced systemic inflammation may be a reduction in BMI in the general population, additive effects of other lifestyle factors such as exercise cannot be excluded. For example, a large body of cross-sectional investigations does indicate that physically active individuals exhibit CRP levels that are 19–35% lower than less active individuals, even when adjusted for BMI as was the case in the current analysis [41]. Further, it's been reported that physical activity at a frequency of as little as 1 day per week is associated with lower CRP in individuals who are otherwise sedentary, while more frequent exercise further reduces inflammation [41].

Significantly, our entire cohort of self-selected apparently healthy individuals did not exhibit clinically high hsCRP, with average BMI also below the overweight thresholds. Because all subjects were voluntarily participating in a personalized wellness platform intended to optimize blood biomarkers that included hsCRP, it is possible that some individuals from across the study groups (both running and sedentary) in our cohort partook in some form of inflammation-reducing dietary and/or lifestyle-based intervention. Thus, that we detected a significant difference in hsCRP between exercising and non-exercising individuals in this self-selected already generally healthy cohort may be suggestive of the potential for additional preventative effect of scheduled physical activity on low-grade systemic inflammation in the generally healthy individual.

**Blood lipids.** Controlled studies that tightly track exercise and the associated adiposity reduction have reported that body fat reduction (and not improvement in fitness as measured via $VO2_{max}$) is a predictor of HDL, LDL, and Tg [46]. Similarly, though BMI is an imperfect measure of adiposity, our 2S-MR analysis suggests that this biomarker is causally associated with improved levels of HDL and Tg, though not LDL. This latter finding replicates a report by Hu et al. who, using the Global Lipids Genetics Consortium GWAS summary statistics, applied a network MR approach that revealed causal associations between BMI and blood lipids, where Tg and HDL, but not LDL, were found to trend toward unhealthy levels with increasing adiposity [47]. On the other hand, others implemented a robust BMI genetic risk score and demonstrated a causal association of adiposity with peripheral artery disease and a multiple linear regression showed a strong association with HDL, TC, and LDL, among other metabolic parameters [48]. In our cohort, given the lack of evidence for a causal BMI-LDL association and the overall healthy levels of BMI, the observed a significant improvement in LDL may be a result of marked running intensity and/or volume, possibly combined with the aforementioned additional wellness program intervention variables.

**Hormonal traits.** As described above, we observed a trend toward increased plasma cortisol and SHBG in runners, particularly PRO level athletes. The effects on cortisol are consistent with a report by Houmard et al., who found male distance runners to exhibit higher levels of baseline cortisol [49]. With respect to the effects of BMI on baseline cortisol levels, this observation is generally supported by our 2S-MR analyses with evidence for a consistent effect of increased cortisol with decreasing BMI. However, this association was suggestive at best, indicating that the higher levels of cortisol exhibited in the PRO runners with significant lower adiposity are not likely to be solely explained by their lower BMI. Indeed, the relationship between BMI and cortisol appears to be complex, with some reports suggesting a U-shaped relationship, where the glucocorticoid's levels associate negatively up to about a BMI of 30 kg/m2, then exhibiting a positive correlation into obesity phenotypes [50]. MR statistical models generally do not account for such non-linearity and would require a more granular demographical treatment, which is not possible using only GWAS summary statistics data in the context of 2S-MR [17, 51].

## Behavioral traits associated with increase physical activity

The combination of the body of the literature that describes the effects of endurance training on blood biomarkers, and our own analysis that included markers such as CK and AST, makes us cautiously assured that most of the abovementioned blood biomarker signatures are indeed a result of the interplay between self-reported running and the associated lower BMI. However, as this is a self-report-based analysis and we were unable to track other subject behaviors in this free-living cohort, we acknowledge that multiple behaviors that associate with exercise may be influencing our results.

Toward this end, our exploratory 2S-MR analyses revealed potentially causal relationships between vigorous exercise and multiple dietary habits that have been shown to improve the biomarkers we examined. Indeed, diets that avoid processed meat and sweets while providing ample amounts of fresh fruits, as well as oily fish have been shown to be anti-inflammatory, and improve glycemic control and dyslipidemia [52, 53]. That physically active individuals are also more likely to make healthier dietary choices adds insight to the potential confounders in ours and others' observational analyses, and this similar associations have previously been reported [54–56]. For example, using a calculated healthy eating motivation score, Naughton et al. showed that those who partake in more than 2 hours of vigorous physical activity are almost twice as likely to be motivated to eat healthy [56]. Indeed, upon closer examination, the genetic instruments used to approximate vigorous physical activity as the exposure in this work included variants in the genes *DPY19L1*, *CADM2*, *CTBP2*, *EXOC4*, and *FOXO3* [57]. Of these, *CADM2* encodes proteins that are involved in neurotransmission in brain regions well known for their involvement in executive function, including motivation, impulse regulation and self-control [58]. Moreover, variants within this locus have been associated with obesity-related traits [59]. Thus, it is likely that the improved metabolic outcomes seen here with our self-reported runners are a composite result of both these individuals exercise and dietary habits. Importantly, the above suggests that a holistic wellness lifestyle approach is in practice the most likely to be most effective toward preventing cardiometabolic disease. Nonetheless, the focus of this work–exercise in the form of running–is known to significantly improve cardiorespiratory fitness (CRF), which has been shown to be an independent predictor of CVD risk and total mortality, outcomes that indeed correlate with dysregulated levels in many of the blood biomarkers examined in this work [7].

## Study limitations

This study is based on self-reported running and thus has several limitations. First, it is generally known that subjects tend to overestimate their commitment to exercise when self-reporting, although in our cohort is a self-selected health-oriented population that is possibly less likely to over-report their running volume. Furthermore, although the robust increasing trend in baselines for muscle damage biomarkers (CK, AST) that have been shown to be associated with participation in sports and exercise provides indirect evidence that the running groups were indeed participating in increasing volumes of strenuous physical activity, we cannot confirm whether the reported running was performed overground or on a treadmill, which may result in some heterogeneity in physiological responses, nor can we ascertain the actual training volume of PRO-level runners. We also cannot exclude the possibility that the running groups also participated in other forms of exercise (such as strength training) or partook in other wellness program interventions that may have influenced their blood biomarkers and/or BMI via lean muscle accretion. Toward this end, we have attempted to shed light on potential behavioral covariates related to vigorous physical activity via 2S-MR. Finally, while this cohort

is generally healthy, we cannot exclude the potential for unmeasured confounders such as medications, nutritional supplements, and unreported health conditions.

2S- MR enables the assessment of causal relationships between modifiable traits and is less prone to the so-called "winner's curse" that more readily affects one-sample MR analyses [17, 51]. Because 2S-MR uses GWAS summary statistics for both exposure and outcome, it is possible to increase statistical power because of the increased sample sizes. However, horizontal pleiotropy is still a concern that can skew the results. Currently, there is no gold standard MR analysis method, thus we used different techniques (IVW, MR-Egger, and median-based estimations–all of which are based on different assumptions and thus biases) to evaluate the consistency among these estimators and only reported associations as 'causal' if there was cross-model consistency. Nonetheless, an exposure such as BMI is a complex trait that is composed of multiple sub-phenotypes (such as years of education) that could be driving the causal associations.

## Conclusions

Running is one of the most common forms of vigorous exercise practiced globally, thus making it a compelling target of research studies toward understanding its applicability in chronic disease prevention. Our cross-sectional study offers insight into the biomarker signatures of self-reported running in generally healthy individuals that suggest improved insulin sensitivity, blood lipid metabolism, and systemic inflammation. Furthermore, using 2S-MR in independent datasets we provide additional evidence that some biomarkers are readily modified BMI alone, while others appear to respond to the combination of varying exercise and BMI. Our additional bi-directional 2S-MR analyses toward understanding the causal relationships between partaking in vigorous physical activity and other healthy behaviors highlight the inherent challenge in disambiguating exercise intervention effects in cross sectional studies of free-living populations, where healthy behaviors such as exercising and healthy dietary habits co-occur. Overall, our analysis shows that the differences between those who run and the sedentary in our cohort are likely a combination of the specific physiological effects of exercise, the associated changes in BMI, and lifestyle habits associated with those who exercise, such as food choices and baseline activity level. Looking ahead, the InsideTracker database is continuously augmented with blood chemistry, genotyping, and activity tracker data, facilitating further investigation of the effects of various exercise modalities on phenotypes related to healthspan, including longitudinal analyses and more granular dose-response dynamics.

## Supporting information

**S1 File.**
(PDF)

**S1 Dataset.**
(TXT)

## Acknowledgments

We thank Michelle Cawley and Renee Deehan for their assistance with background subject matter research and insightful conversations.

## Author Contributions

**Conceptualization:** Bartek Nogal.

**Data curation:** Svetlana Vinogradova.

**Formal analysis:** Bartek Nogal, Svetlana Vinogradova, Paul Fabian.

**Investigation:** Bartek Nogal, Svetlana Vinogradova.

**Methodology:** Bartek Nogal, Svetlana Vinogradova.

**Project administration:** Bartek Nogal.

**Supervision:** Milena Jorge, Ali Torkamani, Gil Blander.

**Visualization:** Svetlana Vinogradova.

**Writing – original draft:** Bartek Nogal.

**Writing – review & editing:** Bartek Nogal.

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
