## [Decision Letter · Decision Letter 0]

13 Sep 2023

PONE-D-23-25168Dose response of running on blood biomarkers of wellness in the generally healthy PLOS ONE

Dear Dr. Nogal,

Thank you for submitting your manuscript to PLOS ONE. After careful consideration, we feel that it has merit but does not fully meet PLOS ONE’s publication criteria as it currently stands. Therefore, we invite you to submit a revised version of the manuscript that addresses the points raised during the review process.

Please submit your revised manuscript by Nov 27 2023 11:59PM. If you will need more time than this to complete your revisions, please reply to this message or contact the journal office at plosone@plos.org. Please include the following items when submitting your revised manuscript:A rebuttal letter that responds to each point raised by the academic editor and reviewer(s). You should upload this letter as a separate file labeled 'Response to Reviewers'.A marked-up copy of your manuscript that highlights changes made to the original version. You should upload this as a separate file labeled 'Revised Manuscript with Track Changes'.An unmarked version of your revised paper without tracked changes. You should upload this as a separate file labeled 'Manuscript'.

We look forward to receiving your revised manuscript.

Kind regards,

Efrem Kentiba, PhD

Academic Editor

PLOS ONE

Journal Requirements:

2.We note that the grant information you provided in the ‘Funding Information’ and ‘Financial Disclosure’ sections do not match. 

3.Thank you for stating the following in the Acknowledgments Section of your manuscript: 

InsideTracker is the sole funding source.  We thank Michelle Cawley and Renee Deehan for their assistance with background subject matter research and insightful conversations.

InsideTracker was the sole funding source.

4. Thank you for providing the following Funding Statement:  

B.N., S.V., P.F., and G.B. are employees of InsideTracker.

We note that one or more of the authors is affiliated with the funding organization, indicating the funder may have had some role in the design, data collection, analysis or preparation of your manuscript for publication; in other words, the funder played an indirect role through the participation of the co-authors. 

If the funding organization did not play a role in the study design, data collection and analysis, decision to publish, or preparation of the manuscript and only provided financial support in the form of authors' salaries and/or research materials, please review your statements relating to the author contributions, and ensure you have specifically and accurately indicated the role(s) that these authors had in your study in the Author Contributions section of the online submission form. Please make any necessary amendments directly within this section of the online submission form.  Please also update your Funding Statement to include the following statement: “The funder provided support in the form of salaries for authors [insert relevant initials], but did not have any additional role in the study design, data collection and analysis, decision to publish, or preparation of the manuscript. The specific roles of these authors are articulated in the ‘author contributions’ section.” 

If the funding organization did have an additional role, please state and explain that role within your Funding Statement. 

Please also provide an updated Competing Interests Statement declaring this commercial affiliation along with any other relevant declarations relating to employment, consultancy, patents, products in development, or marketed products, etc.  

6.We note that you have included the phrase “data not shown” in your manuscript. Unfortunately, this does not meet our data sharing requirements. PLOS does not permit references to inaccessible data. We require that authors provide all relevant data within the paper, Supporting Information files, or in an acceptable, public repository. Please add a citation to support this phrase or upload the data that corresponds with these findings to a stable repository (such as Figshare or Dryad) and provide and URLs, DOIs, or accession numbers that may be used to access these data. Or, if the data are not a core part of the research being presented in your study, we ask that you remove the phrase that refers to these data.

7.Please include captions for your Supporting Information files at the end of your manuscript, and update any in-text citations to match accordingly. Please see our Supporting Information guidelines for more information: http://journals.plos.org/plosone/s/supporting-information. 

8. When submitting your revision, we need you to address these additional requirements. Please ensure that your manuscript meets PLOS ONE's style requirements, including those for file naming. The PLOS ONE style templates can be found at: https://journals.plos.org/plosone/s/submission-guidelines. 

Reviewers' comments:

Reviewer's Responses to Questions

**Comments to the Author**

1. Is the manuscript technically sound, and do the data support the conclusions?

Reviewer #1: Partly

Reviewer #2: Yes

Reviewer #3: Yes

2. Has the statistical analysis been performed appropriately and rigorously? 

Reviewer #1: Yes

Reviewer #2: Yes

Reviewer #3: Yes

3. Have the authors made all data underlying the findings in their manuscript fully available?

Reviewer #1: Yes

Reviewer #2: Yes

Reviewer #3: Yes

4. Is the manuscript presented in an intelligible fashion and written in standard English?

Reviewer #1: Yes

Reviewer #2: Yes

Reviewer #3: Yes

5. Review Comments to the Author

Reviewer #1: The majority parts of the articles are technically sound. Moreover, the purpose of the study is very sound since it focused on healthy active population. Among the few drawbacks of the study the way the study subjects categorized into groups based on the duration of the activity (>10hours per week and less that 10hours per week is not appropriate. Moreover, the reliability/validity of the information sources in relation to the biomarker tests and lifestyle habits of the study subjects didn't consider the immediate effects of medical services and medication conditions of the respondents at the time of reporting the volume of exercise and biomarker test results. Medical services and lifestyle habits specially all the habits in addition to exercises/running are very important to reach informative decision in this research. So, the above two points need further explanation or modification.

Reviewer #2: Comments:

How your data is reliable by using A cross-sectional study design? &

How again the Data is reliable by using self-reported running. I understand that Biomarkers are objective measure, but do you think that Self-report is trustworthy?

Thank you

Reviewer #3: Upon a meticulous review of the article in question, I wish to commend the authors for crafting a piece that not only carries immense scientific weight but is also articulated with great clarity. Such insightful work surely merits publication in your distinguished journal. It's admirable how the authors have navigated through a myriad of physiological and biochemical variables (blood biomarkers) across five distinct participant categories and presented their results with lucidity. The experimental framework is robust, the statistical evaluations are apt, and the narrative progresses seamlessly. The references provided are both relevant and adequate. Nevertheless, I'd like to offer a few observations and suggestions:

Original Title: “Dose response of running on blood biomarkers of wellness in the generally healthy.”

Proposed Title: “Dose-response relationship between running and blood biomarkers of wellness in generally healthy individuals.”

Page 2, Line 8: The mention of “exposure to sunlight” seems somewhat out of context. Could the authors clarify its relevance or indicate if it has been discussed elsewhere in the article?

Page 17, Lines 17-18: The text reads: "These observations suggest that elite endurance runners………to their magnesium status."

Comments: It would be helpful to clarify whether the professional athletes (PRO) participating in this study are specifically elite endurance runners. Kindly integrate this distinction into the main text if accurate.

Page 19, Lines 1-2: The assertion: “Indeed whether exercise………..is inconclusive,” needs to be substantiated with a relevant citation.

Table 1: Please include standard deviation (SD) values. I also recommend expressing exercise duration in terms of "h/week" instead of "hr".

6. PLOS authors have the option to publish the peer review history of their article (what does this mean?). If published, this will include your full peer review and any attached files.

Reviewer #1: No

Reviewer #2: No

Reviewer #3: **Yes: **Dr. Subir Gupta

---

## [Author Response · Author response to Decision Letter 0]

3 Oct 2023

Editor comments:

1. When submitting your revision, we need you to address these additional requirements. Please ensure that your manuscript meets PLOS ONE's style requirements, including those for file naming.

Thank you. We believed we now correctly formatted the manuscript to reflect PLOS formatting requirements for publishing.

2.We note that the grant information you provided in the ‘Funding Information’ and ‘Financial Disclosure’ sections do not match. 

Thank you - we believe that we now addressed this. We removed all funding information from the manuscript text, amended it to reflect the funding institutions involvement, and moved it to the cover letter, as requested. Please let us know if anything needs further attention.

 3. In your Data Availability statement, you have not specified where the minimal data set underlying the results described in your manuscript can be found.

We updated our Data Availability statement via the submission system to reflect our opennes to share the minimal dataset upon request from the corresponding author, as well as the url to the public repository where the gwas summary statistics can be found.

4. We note that you have included the phrase “data not shown” in your manuscript....

Thank you. We now addressed this within the text of the manuscript and the “data not shown” no longer appears

5. Please include captions for your Supporting Information files at the end of your manuscript, and update any in-text citations to match accordingly. 

Captions for Supporting Information is now included at the end of the manuscript.

6. When submitting your revision, we need you to address these additional requirements. Please ensure that your manuscript meets PLOS ONE's style requirements, including those for file naming. 

We believe we have now addressed the style and formatting requirements.

Reviewer comments:

Reviewer #1: The majority parts of the articles are technically sound. Moreover, the purpose of the study is very sound since it focused on healthy active population. Among the few drawbacks of the study the way the study subjects categorized into groups based on the duration of the activity (>10hours per week and less that 10hours per week is not appropriate. Moreover, the reliability/validity of the information sources in relation to the biomarker tests and lifestyle habits of the study subjects didn't consider the immediate effects of medical services and medication conditions of the respondents at the time of reporting the volume of exercise and biomarker test results. Medical services and lifestyle habits specially all the habits in addition to exercises/running are very important to reach informative decision in this research. So, the above two points need further explanation or modification.

Response to Reviewer #1: We appreciate the reviewer's feedback and are pleased that they find the majority of our study technically sound and recognize the importance of our focus on a healthy, active population. We also appreciate the reviewer pointing out an opportunity to improve the clarity around our experimental design as it pertains subject groupings.

Regarding the categorization of study subjects, we want to clarify that we actually categorized them into five groups. These groups include professional endurance runners, high volume amateur runners (>10 hours per week), medium volume amateur runners (3-10 hours per week), low volume amateur runners (<3 hours per week), and the sedentary. We now added a sentence starting on line 125 the explicitly states this categorization (“The cohort was divided into five groups:…”). These groupings were determined based on the respondents' self-reported data.

We acknowledge the potential influence of medication use on our analysis, and we now address it starting on line 461 (“These factors, such as diet, sleep, and/or medications were not readily ascertained in this free-living cohort…”) and in the Study Limitations section (line 595). We noted that unmeasured confounders such as medications, nutritional supplements, and unreported health conditions may exist. However, given the nature of our cohort, which primarily consists of self-selected, generally healthy individuals, the impact of significant medication use is expected to be limited. We believe that the observed trends in healthier biomarker levels with increased reported running volume support this assertion.

Furthermore, we recognize the importance of lifestyle habits beyond exercise in influencing our results. To address this, we employed statistical genomics, specifically two-sample Mendelian randomization with physical activity as the exposure. This analysis allowed us to explore other potential habits and behaviors contributing to improved biomarker signatures in physically active runners within our cohort. We kindly refer the reviewer to the "Vigorous physical activity associates with healthier behaviors" section in the results for a detailed examination of this aspect. Notably, our entire cohort is composed of health-conscious individuals within the same health advisory platform, with the primary differentiator being self-reported running activity. We also controlled for key variables such as age, sex, and BMI in our ANOVA analyses.

We hope these explanations clarify our approach and address the reviewer's concerns adequately.

Reviewer #2: How your data is reliable by using A cross-sectional study design?

How again the Data is reliable by using self-reported running. I understand that Biomarkers are objective measure, but do you think that Self-report is trustworthy? Thank you

Response to Reviewer #2: We appreciate the reviewer's questions and concerns regarding the reliability of our runners data, which is largely derived from self-reported exercise habits. Cross-sectional studies inherently have limitations when it comes to establishing causality, and we acknowledge this challenge. To address potential confounding factors, we conducted additional causal analyses, specifically investigating the effects of BMI on the biomarkers under examination to begin to disentangle the relative contributions of known factors. Furthermore, we performed secondary Mendelian randomization (MR) analyses to identify and account for potential confounders in our findings. We kindly invite the reviewer to explore the "Vigorous physical activity associates with healthier behaviors" section in the results for a comprehensive exploration of these confounding aspects.

Regarding the reliability of self-reported running activity, we recognize that self-reports can be subject to biases, and individuals may tend to overestimate their exercise commitment. To address this drawback, we added language addressing these limitations in the “Study limitations” section (Line 579: “First, it is generally known that subjects tend to overestimate their commitment to exercise …”). We do note that our study cohort comprises self-selected individuals who are health-conscious and possibly less prone to over-report their running volume. Additionally, the robust increasing trend in baseline levels of muscle damage biomarkers (CK, AST), which are known to be associated with participation in sports and exercise, provides indirect evidence that the different running groups in our study were indeed engaging in increasing volumes of strenuous physical activity.

While self-reporting has its limitations, it remains a valuable method for capturing individuals' exercise behaviors in large-scale observational studies. We took measures to mitigate potential biases, and our findings align with established trends in biomarker responses to physical activity.

Reviewer #3: Upon a meticulous review of the article in question, I wish to commend the authors for crafting a piece that not only carries immense scientific weight but is also articulated with great clarity. Such insightful work surely merits publication in your distinguished journal. It's admirable how the authors have navigated through a myriad of physiological and biochemical variables (blood biomarkers) across five distinct participant categories and presented their results with lucidity. The experimental framework is robust, the statistical evaluations are apt, and the narrative progresses seamlessly. The references provided are both relevant and adequate. Nevertheless, I'd like to offer a few observations and suggestions:

Response: We appreciate the reviewer's positive feedback and kind words about our manuscript. We eagerly await their observations and suggestions should they see further opportunities to improve our work based on our responses to the current suggestions.

Original Title: “Dose response of running on blood biomarkers of wellness in the generally healthy.”

Proposed Title: “Dose-response relationship between running and blood biomarkers of wellness in generally healthy individuals.”

Response: Thank you – title has been changed.

Page 2, Line 8: The mention of “exposure to sunlight” seems somewhat out of context. Could the authors clarify its relevance or indicate if it has been discussed elsewhere in the article?

Response: Thank you for the suggestion, we removed this as we agree it was not relevant in this manuscript.

Page 17, Lines 17-18: The text reads: "These observations suggest that elite endurance runners………to their magnesium status."

Comments: It would be helpful to clarify whether the professional athletes (PRO) participating in this study are specifically elite endurance runners. Kindly integrate this distinction into the main text if accurate.

Response: Thank you for the clarifying suggestion. We included the pro/elite endurance runners clarification within the abstract as well as a section heading (lines 7 and 425)

Page 19, Lines 1-2: The assertion: “Indeed whether exercise………..is inconclusive,” needs to be substantiated with a relevant citation.

Response: Thank you – citations have been added. 

Table 1: Please include standard deviation (SD) values. I also recommend expressing exercise duration in terms of "h/week" instead of "hr".

Response: Thank you for the catch – units changed to “h/week” and SDs added to Table 1. 

We are grateful for your valuable feedback, which has contributed to improving the clarity and accuracy of our manuscript.

---

## [Decision Letter · Decision Letter 1]

17 Oct 2023

Dose response of running on blood biomarkers of wellness in generally healthy individuals

PONE-D-23-25168R1

Dear Dr. Nogal

We’re pleased to inform you that your manuscript has been judged scientifically suitable for publication and will be formally accepted for publication once it meets all outstanding technical requirements.

Kind regards,

Efrem Kentiba, PhD

Academic Editor

PLOS ONE
---

## [Editor Report · Acceptance letter]

20 Oct 2023

PONE-D-23-25168R1 

Dose response of running on blood biomarkers of wellness in generally healthy individuals 

Dear Dr. Nogal:

I'm pleased to inform you that your manuscript has been deemed suitable for publication in PLOS ONE. Congratulations! Your manuscript is now with our production department. 

Kind regards, 

on behalf of

Dr. Efrem Kentiba 

Academic Editor

PLOS ONE